# Experimental Study and Numerical Analysis of Chloride Ion Diffusion in Hydrotalcite Concrete in Chloride Salt Environment

**DOI:** 10.3390/ma16196349

**Published:** 2023-09-22

**Authors:** Lina Zhou, Ying Cai, Cailong Ma

**Affiliations:** 1School of Civil Engineering and Architecture, Xinjiang University, Urumqi 830047, China; augcy@xju.edu.cn (Y.C.); macailong@xju.edu.cn (C.M.); 2Heilongjiang Province Hydraulic Research Institute, Harbin 100050, China

**Keywords:** LDHs, NT-Build 443, chloride ion diffusion, COMSOL Multiphysics

## Abstract

Hydrotalcite, known as layered double hydroxides (LDHs), is a new type of admixture used to delay the corrosion of reinforcement. The aim of this study was to investigate the chloride ion diffusion behavior of C30 concrete with varying amounts of calcined hydrotalcite (0%, 2%, 4% and 6%) in a chloride salt environment. The NT-Build 443 test was adopted to characterize the one-dimensional accelerated chloride ion penetration of concrete. The distribution of chloride ion concentration in hydrotalcite concrete with different mix proportions immersed in sodium chloride solution for 30 days and 60 days was determined, and the chloride ion diffusion coefficient and surface chloride ion concentration were fitted based on Fick’s second law to establish the chloride ion diffusion model considering the influence of multiple factors. This model was validated using COMSOL Multiphysics finite element software. The results show that concrete mixed with LDHs can meet its compressive strength requirements and that the resistance of concrete with 2% calcined hydrotalcite to chloride ion penetration is the best with a 19.6% increase in the 30-day chloride ion penetration coefficient. The chloride ion diffusion process under chloride salt immersion conditions is in accordance with Fick’s second law. The chloride ion concentrations calculated with COMSOL software and the test results are in good agreement, which verifies the reliability of the chloride ion diffusion model.

## 1. Introduction

As the most widely used material in construction projects, the durability of concrete has received great attention from scholars at home and abroad. Due to the defects and pore structure of concrete, chloride ions in the external environment penetrate into concrete through pores and cracks, and when the chloride ion concentration on the surface of reinforcement reaches a certain threshold, the passivation film is damaged, inducing the corrosion of reinforcement [1], which seriously affects the service life of concrete structures [2,3,4,5]. Concrete structures often suffer from salt freeze damage and salt corrosion damage caused by seawater scouring, especially in the saline soil areas of northwest China and southeastern coastal areas, which not only brings huge economic losses but is also accompanied by safety hazards [6,7].

In recent years, many scholars have conducted extensive research on the diffusion of chloride ions in concrete in chloride salt environments with both experimental studies and numerical simulations and have achieved certain results. Domestic and foreign scholars [8,9,10] have exploited the nature of LDHs to investigate the influence of the LDH structure on the resistance of cementitious materials to chloride ion penetration from the perspective of modification. Tatematsu et al. [11] first proposed that aqueous calcium carbide-like materials could be used as salt sorbents in cementitious materials in 2003, and the following year, Raki et al. [12] demonstrated that aqueous hydrotalcite-like-phase materials incorporated into concrete could control the release rate of organic mixtures. The adsorption of chloride ions by CLDHs incorporated in concrete can increase the chloride ion threshold on the surface of reinforcing steel, thus delaying the corrosion of reinforcing steel. Hu Jing et al. [13] and Zhang Lin [14] conducted a fundamental study on the adsorption mechanism of chloride ions by MgAl-CO_3_ CLDHs based on adsorption thermodynamics in simulated pore liquids. On this basis, Lin Zhang further investigated the chloride fixation capacity and mechanism of CLDHs in cement paste, and the results showed that the chloride fixation capacity of the test group doped with CLDHs was better compared with the control group; it is noteworthy that the chloride fixation capacity of CLDHs showed an increasing trend with the increase in curing age, which was positively correlated with the chloride ion concentration. The chloride ions in concrete can be divided into bound chloride ions and free chloride ions, and it is generally believed that free chloride ions destroy the passivation film at the reinforcing steel interface [15]. Wang Pei [16] tested the chloride ion concentration distribution in cementitious composites mixed with different amounts of CLDHs; the results demonstrated that the admixture of CLDHs fills the pores of cementitious materials, enhances their compactness and increases the resistance to chloride ions, and the chloride fixation capacity increases with the increase in LDH admixture. Based on the ion-exchange function of LDHs, Ping Duan [17], Ma [18] and Shui [19] evaluated the anti-carbonation properties of concrete with CLDHs obtained using different treatment methods. The results of the aforementioned studies showed that the performance of CLDHs in the adsorption of chloride ions was superior to that of LDHs, which was verified by Liu et al. [20].

With the rapid development of computer technology, numerical simulation has been widely used in research on the durability of concrete. The chloride ion diffusion model under the influence of multiple factors is established by firstly simulating the chloride ion diffusion process in the saturated state [21,22] using Fick’s second law and then defining the initial and boundary conditions as well as the parameters of each influencing factor. Paul et al. [23] explored the chloride ion diffusion coefficient and diffusion depth of concrete based on Fick’s second law and COMSOL Multiphysics 6.0 Multiphysics for numerical simulation, considering factors such as time, temperature, chloride ion binding action and the mix ratio of concrete. A numerical solution was also given for the chloride ion transportation model under multiphysics field coupling using COMSOL Multiphysics software [24]. In addition, a chloride ion diffusion–convection model was established based on Fick’s second law and Darcy’s law, and COMSOL Multiphysics was adopted to realize the numerical simulation of the multiphysics field coupling to effectively characterize the diffusion law of chloride ions in concrete [25].

Therefore, this study focuses on the diffusion characteristics and laws of chloride ions in hydrotalcite concrete in a chloride salt environment and establishes a multi-factor-influenced chloride ion diffusion model. Based on a COMSOL simulation analysis of the chloride ion diffusion model and a comparison with experimental data, the chloride ion diffusion model of hydrotalcite concrete in a chloride salt environment is gradually improved.

## 2. Materials and Methods

### 2.1. Materials and Specimen Preparation

P.O 42.5R Portland cement produced by Urumqi Tianshan Cement Plant in accordance with Chinese standard GB175-2007, with compressive strength of 51.1 MPa at the age of 28 days, was used for the preparation of concrete. Layered double hydrotalcite (LDH for short) was obtained from Shandong New Weiner Material Technology Co., Ltd., Weifang, China, and its chemical composition is given in Table 1. To obtain a good chloride fixation ability, the LDHs should be calcinated at 500 °C for 4 h and its structure changed, which led to the formation of LDOs. And the dosage of LDOs was 2%, 4% and 6%, substituting the same amount of cement. The coarse aggregates were the crushed granite with a maximum size of 20 mm and bulk density of 2650 kg/m^3^ from the Xinjiang region. River sand with a fineness modulus of 2.38 was used as the fine aggregate. The total binding materials was 346 kg/m^3^, and the w/b was 0.45. A superplasticizer was used, and its dosage was adjusted to keep the slump of fresh concrete between the 140 mm and 160 mm range. The mix proportions, conforming to Chinese specification for the mix proportion design of ordinary concrete JGJ55-2011 [26], and the corresponding compressive strength and chloride ion permeability coefficient of concrete cured in a standard room (T = 20 ± 2 °C, R5%) are given in Table 2.

According to the Standard for Test Methods of Physical and Mechanical Properties of Concrete (GB/T 50081-2019) [27], hydrotalcite concrete specimens with size of 100 mm × 100 mm × 100 mm were prepared for measurements. Cylindrical specimens with a diameter of 100 mm and a height of 50 mm were prepared to monitor chloride transportation. And after hardening for 24 h, specimens were demolded and cured at 20 ± 2 °C and 95% relative humidity for 7 days, 14 days, 28 days and 56 days, respectively.

### 2.2. Hydrotalcite Chloride Ion Fixation Test

The LDHs and LDOs materials before and after roasting were added to a 1% mass fraction of NaCl solution in the ratio of 1:100, stirred well, and then placed on the hydrothermal device to fully react for 24 h. After the reaction was completed, the solution was extracted and filtered, and the filter residue was loaded in a quartz crucible and placed in an oven at 60 °C for 24 h. Finally, XRD and SEM were conducted to characterize the chloride fixation ability and mechanism.

### 2.3. Compressive Strength Test

In order to evaluate the effect of LDOs on the compressive strength of concrete, a microcomputer-controlled electro-hydraulic servo pressure tester was used to test the compressive strength of hydrotalcite concrete at 7 days, 14 days, 28 days and 56 days. Each compressive strength was the average value of three measured ones.

### 2.4. One Dimensional Accelerated Chloride Penetration Test

The one-dimensional diffusion of chloride ions in concrete specimens was studied, which was complaint with the NT-Build 443 standard. To prevent the interference of multi-dimensional diffusion situation, it is necessary to use paraffin wax to seal around the test block, and it is sufficient to keep two opposite sides.

The surface-treated specimen was immersed in saturated Ca(OH)_2_ solution until the mass of the specimen in the surface dry state no longer changed; then, it was immersed in NaCl solution with a concentration of 2.82 mol/L for 30 days and 60 days, and we covered the immersion basin with plastic film to prevent the concentration from increasing due to excessive water evaporation. During the soaking period, the concentration of the solution was tested every 10 days, and if the concentration of NaCl solution increased, an appropriate amount of distilled water was added to keep the concentration fixed at 2.82 mol/L.

The specimen blocks after completed soaking were taken out and placed at room temperature for 24 h and then sampled. The core samples were drilled from 5 different depths of the test blocks (0~5 mm, 5~10 mm, 10~15 mm, 15~20 mm, 20~25 mm, respectively). Figure 1 shows the sampling diagram of the specimen block. The powder samples were obtained over a 0.63 mm sieve; the sieve residual samples were drying in an oven at 40 °C for 24 h until completely dry and then cooled to room temperature in a dry and clean vessel. Finally, the chloride ion concentration in the powder samples was tested by potentiometric titration, and the chloride ion concentration–depth curves were plotted and analyzed by nonlinear regression using Fick’s second law.

### 2.5. Chloride Ion Concentration Measurement

According to Chinese methods for testing the uniformity of concrete admixtures (GB/T8077-2012) [28], weigh each 5 g sample to an accuracy up to 0.0001 g. Pour it into a 250 mL beaker, add 20 mL of distilled water for stirring to make the sample completely dispersed, and add 25 mL nitric acid (1 + 1) under stirring so that the sample does not contain particles that are not completely reacted. Then, add 100 mL distilled water to dilute the solution and add 2 mL chloride ion standard solution and 2 mL hydrogen peroxide dropwise, cover the surface dish, boil the solution by heating for 1~2 min and then cool to room temperature. Put 1~2 magnetic stirrers in the beaker containing the sample solution and place it on the automatic potentiometric titrator to start titration until the end point.

### 2.6. Microscopic Test

X-ray diffraction analysis samples of the control REF and L2 group were taken before and immersed in the chloride salt immersion for 56 days. The removed sample powder was sieved and then soaked in anhydrous ethanol to terminate the hydration; then, it was placed in an oven at a constant temperature of 60 °C for 24 h before the test.

A scanning electron microscope was used to observe the morphology features of specimens before and after soaking in NaCl solution, and the sample size was 3~5 mm in diameter and 3~5 mm in thickness, which was subjected to vacuum treatment and gold spraying before scanning so that the microscopic morphology could be analyzed and tested.

### 2.7. Statistical Method of Parameter Calculation

#### 2.7.1. Fick’s Second Law

The common modes of chloride ion transport in concrete are mainly diffusion, pressure penetration, capillary absorption, and charge migration. In the saturated state, the chloride ion transport is mainly by diffusion. While in the unsaturated state, the chloride ion migration process is influenced by diffusion and capillary adsorption [29]. The diffusion process of chloride ions in concrete can be illustrated by Fick’s second law.
(1)$$\frac{𝜕C(x,t)}{𝜕t}=\frac{𝜕}{𝜕x}[D\times \frac{𝜕C(x,t)}{𝜕x}]$$
where *t* is the time, s; *x* is the distance, m; and *D* is the diffusion coefficient, m^2^/s.

Set the initial condition *C*(*x*, 0) = *C*_0_, the boundary condition *C*(0, *t*) = *C_s_*, and obtain the analytical solution of Fick’s second law [30].
(2)$$C\left(x,t\right)=C_{0}+\left(C_{S}-C_{0}\right)\left[1-erf\left(\frac{x}{2\sqrt{Dt}}\right)\right]$$
where *C*_0_ is the initial chloride ion concentration, and *C_s_* is the surface chloride ion concentration, %.

#### 2.7.2. Time Decay Coefficient

The internal pore structure of concrete provides a pathway for the chloride ion diffusion in concrete; therefore, the compactness of concrete and the diffusion coefficient of chloride ions are closely related. As the curing age increases, the hydration process of cement advances, continuously generating hydration products filling the internal pores and making the concrete structure more dense, which in turn reduces the chloride ion diffusion coefficient. This indicates that the chloride ion diffusion coefficient tends to decrease gradually with increasing curing age, and Thomas et al. [31] proposed the exponential decay equation:(3)$$D(t)=D_{ref}{\left(\frac{t_{ref}}{t}\right)}^{m}$$
where *t_ref_* is the reference curing age, 28 days, *t* is the erosion time of concrete suffered from chloride ions, day; *D_ref_* and *D* represent the corresponding chloride ion diffusion coefficient at the specified time *t_ref_* and *t*, m^2^/s; and *m* is the time decay coefficient.

#### 2.7.3. Concrete Deterioration Factor

Concrete is a complex multi-phase composite material, subjected to long-term effects of humidity, temperature, and stress loading in the service environment, which can lead to the internal structure damage in the concrete, thus providing a pathway for the diffusion of chloride ions, and the chloride ion diffusion coefficient increases subsequently. Weijun et al. [32] proposed the equation of the relationship between the deterioration coefficient of concrete and the diffusion coefficient of chloride ions under different water–cement ratios.
(4)$$M_{d}=\left\{\begin{array}{ll} {}[1000{\left(w/c\right)}^{2}-1050(w/c)+287]/3 & \left(w/c\right)<0.5\\ 4 & \left(w/c\right)\geq 0.5 \end{array}\right.$$
where *M_d_* is the concrete deterioration coefficient, and *w*/*c* is the water–cement ratio of concrete.

#### 2.7.4. Chloride Ion-Binding Capacity

Chloride ions in concrete exist mainly in two forms, where C-S-H gels and C_3_A solidify chloride ions by physisorption and chemical binding, respectively [33]. Sergi [34] proposed a relationship between total chloride ion content *C_t_*, free chloride ion content *C_f_*, and bound chloride ion content *C_b_*, as shown in Equation (5). Sergi considered that the effect of concrete porosity on the concrete chloride ion diffusion coefficient needs to be taken into account and proposed a coefficient $\omega_{e}$, which is defined as the ratio of the volume of evaporable pore water in concrete to the volume of concrete, which is generally approximated as the porosity and taken as 8%.
(5)$$C_{t}=C_{b}+\omega_{e}C_{f}$$

Nilsson et al. [35] defined the chloride-binding capacity *R*, as shown in Equation (6), which was generally measured through a test.
(6)$$R=\frac{1}{\omega_{e}}\times \frac{𝜕C_{b}}{𝜕C_{f}}$$

The coefficient of influence of chloride-binding capacity *M_R_* can be obtained through solving Fick’s second law equation, taking into account the coefficient of chloride-binding action *R*, as shown in Equation (7).
(7)$$M_{R}=\frac{1}{1+R}$$

#### 2.7.5. Internal Relative Humidity Influence Factor

The significant effect of internal relative humidity on chloride ion diffusion is due to the fact that moisture, as a carrier of chloride ion diffusion, can provide channels for chloride ion diffusion in higher humidity environments and accelerate the chloride ion diffusion process. Conversely, it decreases the chloride ion diffusion coefficient. Bazant et al. [36] proposed an equation between the chloride ion diffusion coefficient and internal relative humidity, as shown in Equation (8).
(8)$$M_{RH}={\left[1+\frac{{\left(1-RH\right)}^{4}}{{\left(1-RH_{C}\right)}^{4}}\right]}^{-1}$$
where *RH* is the internal relative humidity of concrete, and *RH_C_* is the critical relative humidity of concrete, 0.75.

#### 2.7.6. Temperature Parameters

The chloride ion diffusion coefficient is affected by temperature mainly in terms of water evaporation and cement hydration. On the one hand, water, as a carrier of chloride ion migration and diffusion in concrete, will evaporate with the increase in temperature, thus reducing the chloride ion diffusion coefficient. In addition, the cement hydration process advances with the increase in temperature, and the generated hydration products fill the internal pores of concrete, which in turn reduces the porosity, improves the concrete compactness and slows down the chloride ion diffusion process. On the other hand, the increase in temperature makes the ion movement more active and the diffusion coefficient increases with the increase in ionic activity. The relationship between temperature and chloride ion diffusion coefficient is described by the Arrhenius equation [37], as shown in Equation (9).
(9)$$M_{T}=e^{\frac{U}{R}(\frac{1}{T_{ref}}-\frac{1}{T})}$$
where *R* is the molar gas constant, 8.314 J/(mol·K); *T_ref_* and *T* are the ambient temperature of concrete at the specified time *t_ref_* and *t*, K; and *U* is the ion activation energy, KJ/mol, which is related with the mix proportion of concrete.

#### 2.7.7. Multi-Factor Chloride Ions Diffusion Model

The multi-factor chloride ions diffusion model in this paper was built through key parameters modification based on tested experimental data and models in references [29,38,39], as shown in Equation (10).
(10)$$D=D_{ref}M_{t}M_{d}M_{R}M_{RH}M_{T}$$
where *D_ref_* means the diffusion coefficient at reference time, m^2^/s. *D_ref_* and *C_s_* can be fitted through Origin software based on an analytical solution of Fick’s second law, as shown in Table 3. The immersion time is short and there is no load, so the value of *M_d_* is 1.0. For common concrete, take the coefficient of chloride-binding action *R* as 3, and then, the chloride ion-binding capacity *M_R_* was 0.25, which was calculated according to Equation (7). Taking the internal relative humidity of concrete as 0.9, the calculated *M_RH_* was 0.98. The ambient temperature of concrete was 294.15 K, so the temperature-affected parameters *M_T_* is 1.0.

## 3. Test Results

### 3.1. Compressive Strength of Hydrotalcite Concrete

Figure 2 shows the compressive strength of concrete with different amounts of hydrotalcite at different curing ages. It can be seen from Figure 2 that with the increase in curing age, the compressive strength of each group of concrete showed an increasing trend, and at the same age, the compressive strength first increased and then decreased, the inflection point occurring at 2%. The 28-day compressive strength of each group of concrete met the design strength, and the strength of concrete added with 2% and 4% hydrotalcite increased by 10.95% and 4.23% compared with the control group, respectively. When 6% hydrotalcite was added, its strength decreased by 1.24% compared with the control group, because an excessive amount would inhibit the hydration process of cement and was not conducive to the development of strength.

### 3.2. One-Dimensional Accelerated Chloride Penetration

Figure 3 shows the tested and fitted chloride ion concentration distribution curves of chloride ion concentration in concrete with different LDOs admixtures. It can be concluded that the chloride ion concentration distribution shows a decreasing trend with the increase in depth, and the chloride ion concentration increased firstly and then decreased with the increase in hydrotalcite dosages. In the chloride salt environment, chloride ions diffused from the surface to the interior of the concrete through the water, and the chloride ion concentration of the concrete in the REF group was significantly higher than that of the concrete mixed with hydrotalcite. In addition, the chloride ion concentration inside the concrete showed an increasing trend with the increase in soaking time; compared with the soaking time of 30 days, the chloride ion concentration inside the concrete with soaking time of 60 days generally increased by more than 10%, and the surface chloride ion concentration was significantly larger than the chloride ion concentration inside the concrete. This can be explained by the following two reasons: on one hand, the appropriate amount of hydrotalcite can optimize the pore structure and enhance the compactness of concrete; on the other hand, hydrotalcite can use its interlayer ion exchange properties to complete the adsorption of chloride ions.

Under the chloride salt immersion environment, the chloride ion diffusion inside the concrete is in accordance with Fick’s second law, and the chloride ions migrating into the interior concrete during the moisture transport process carry out a secondary hydration reaction with the incompletely hydrated cementitious material. The rehydration reaction can change the solution concentration inside the concrete on the one hand, and they can further improve the internal pore structure of concrete through cement hydration on the other hand. Therefore, to further study the distribution law of chloride ions in concrete with different LDOs admixtures, the experimental data can be fitted by the analytical solution of Fick’s second law, as shown in Equation (2), using Origin Pro analysis software to obtain the diffusion coefficient D and the surface chloride ion concentration Cs. The fitting results are shown in Table 3 for the internal chloride ion of hydrotalcite concrete under a chloride salt environment distribution fitting curve. From the table, it can be found that when at the same soaking time, the diffusion coefficient D and surface chloride ion concentration Cs gradually decreased with the increase in LDOs dosages, and with the increase in soaking time, the diffusion coefficient of the same group of specimens gradually decreased, while the surface chloride ion concentration gradually increased.

### 3.3. Chloride Adsorption Mechanism of LDOs

#### 3.3.1. XRD

The raw LDHs have a unique layered structure with anion exchange, structural reconstruction, and memory effect properties that make the baked LDOs widely used in ion adsorption. Using this property, the adsorption of chloride ions in concrete was accomplished, and microscopic means were taken to characterize the chloride ion adsorption performance of LDO materials in concrete.

Figure 4 shows the XRD diffraction spectra of the specimens of the REF group and L2 group before and after immersion in chlorine salt solution for 56 days. As can be seen from the figure, the diffraction spectra of the control REF and L2 specimens before and after chlorine salt immersion are approximately the same, and cement hydration products such as calcium hydroxide, C-S-H gel, and Friedel’s salt can be observed, indicating that the incorporation of LDOs materials does not affect the types of cement hydration products generated. At the same time, it can be seen from the plots of the L2 group specimens in Figure 4b that the intensity of the characteristic diffraction peak of calcium hydroxide becomes weaker compared to the control REF, which indicates that the generation of calcium hydroxide in the cement matrix is reduced due to the substitution of LDO material as a non-cementitious material in equal amounts of cement dosage, which has an effect on the generation of cement hydration products. The distinct diffraction peaks characteristic of Friedel salts were observed in the diffraction patterns of L2 specimens soaked with chloride salts, which indicated that some of the free chloride ions were solidified in the Friedel salts, which in turn improved the chloride ion attack resistance of the concrete. In addition, the characteristic diffraction peaks of MgAl-Cl LDHs were also evident, which indicated that the LDOs material could adsorb the chloride ions penetrated into the concrete pore interior by interlayer anion exchange ability under the chloride salt environment, and it could complete the structural reconstruction while improving the chloride salt attack resistance of concrete.

#### 3.3.2. SEM

Figure 5 depicts the SEM microscopic morphology of concrete specimens from REF and L2 groups before and after chloride salt immersion. Figure 5a shows the microscopic morphology of the control group before REF immersion, from which it can be seen that the pure cement specimen group generated more obvious internal hexagonal flakes of calcium hydroxide, as well as a small amount of needle-like calcium alumina, but its internal structure is more loose. Compared with the concrete specimen of the L2 group mixed with 2% LDOs material, as shown in Figure 5c, its internal flake accumulation, as well as the denseness and integrity of the structure has improved. This indicates that the smaller particle size of LDOs material can exert the microaggregate effect to fill the pores inside the concrete, thus improving the macroscopic properties of the concrete. Figure 5b shows the microscopic morphology of the control REF by chlorine salt immersion, from which it can be seen that the internal structure of the specimen after immersion produces hexagonal lamellar Friedel salt crystals and honeycomb C-S-H gel; Figure 5d shows the microscopic morphology of the L2 concrete specimen after 56-day immersion in chlorine salt solution; at this time, the internal lamellar accumulation is still obvious, which is due to the interlayer anion exchange of the incorporated LDOs material. This is because the incorporated LDOs material can complete the structural reconstruction and restore the lamellar morphology through the interlayer anion exchange functional property; in addition, the chloride ions take water as the carrier and diffuse into the concrete interior through the concrete pore channel as the pathway, and they react with the hydration products to generate hexagonal lamellar Friedel salt crystals. Therefore, the incorporation of 2% LDOs material can improve the chloride ion penetration resistance of concrete specimens.

## 4. Numerical Simulation

### 4.1. Module Selection

Enable the COMSOL software to enter the model wizard window, select the three-dimensional space dimension and then enter the physical field selection module, select the dilute matter transfer module, which can be applied to the matter transfer process in both convection–diffusion as well as diffusion cases, and the basic equation for its solution is the extended Fick’s diffusion equation. Click on add and define the general study as transient for the study of the variables within the physical field over time.

### 4.2. Parameter Setting

Enter into the main interface of operation, set the geometry and material in Component 1 in the Model Developer window, respectively, set the size as 100 mm × 100 mm × 100 mm cube, and then click Create Object to build the 3D concrete finite element model. Since concrete is a porous material, select “Celluar concrete” (i.e., porous concrete) as the built-in material in the material library to define the material and set the basic properties of the material. The three-dimensional cube model established in this study has six boundaries, and in order to simulate the one-dimensional diffusion of the actual test, two relative surfaces need to be retained as the concentration boundary and the flux boundary, and the remaining four boundaries need to be retained as the flux-free boundary, which is a rendering of the boundary of the three-dimensional concrete model. Before simulating the diffusion process of chloride ions in concrete, the three-dimensional concrete finite element model needs to be meshed. The mesh division is closely related to the calculation accuracy of the model, and the appropriate mesh cell size can be selected according to the actual demand. Generally speaking, the finer the mesh division, the larger the computational volume and the longer the computational time. In this study, the conventional cell size is chosen for the model calculation, as shown in Figure 6, for the mesh generation of the 3D concrete model. One-dimensional (1D) chloride ions concentration and the chloride ions diffusion depth from the concrete surface were simulated, and only two opposite sides were kept as described in Section 2.4.

### 4.3. Analysis of Numerical Simulation Results

#### 4.3.1. Effect of Erosion Age on the Distribution of Chloride Ion Concentration

Figure 7 shows the results of chloride ion concentration in REF group specimens with erosion time. The parameters of the REF test group were selected for simulation, and its chloride ion diffusion coefficient was 7.35 × 10^−13^ m^2^/s. When the number of simulation days reached 30 days, 60 days, 90 days and 120 days, the chloride ion diffusion depth was 24.25 mm, 26.25 mm, 28.00 mm and 29.75 mm, respectively. It can be seen from the figure that with the increase in erosion time, the chloride ions keep diffusing into the concrete, and its concentration is decreasing.

#### 4.3.2. Effect of LDOs Dosages on the Distribution of Chloride Ion Concentration

The results of the variation of chloride ion concentration in concrete specimens with LDOs are shown in Figure 8. The chloride ion attack time was determined as 30 days, and the chloride ion distribution in the concrete was simulated by changing the chloride ion diffusion coefficient with different hydrotalcite contents. When mixed amounts of LDOs are 0%, 2%, 4% and 6%, the chloride ion diffusion depth is 24.25 mm, 20.00 mm, 20.45 mm and 21.25 mm, respectively. As can be seen from Figure 8, with the increase in LDO dosages, the chloride ion diffusion depth increased first and then decreased, which is attributed to the appropriate amount of hydrotalcite being able to fill the small pores in the concrete to a certain extent, and it can absorb part of the water in the cement matrix through the interlayer water storage and physical adsorption function, thus reducing the water–cement ratio of concrete, increasing the compactness of concrete and impeding the channel of chloride ion diffusion to the concrete interior. In addition, a non-cementitious material and excessive incorporation of hydrotalcite will hinder the hydration process of cement, which is not conducive to the development of concrete strength, resulting in the reduction in concrete resistance to chloride ion attack performance.

#### 4.3.3. Comparison between COMSOL Numerical Solution and Experimental Data

In order to verify the reliability of the established chloride ion diffusion model by COMSOL software, the measured chloride ion concentration at different depths from the surface of the control group REF and L2 group were selected to compare with the numerical results, and Figure 9 shows the comparison between the numerical results of COMSOL and the test results. As can be seen from the figure, the error is only about 5% at the depth of 5 mm, which indicates that the established chloride ion diffusion model can effectively reflect the distribution of chloride ion concentration inside the test concrete.

## 5. Conclusions

The improvement of LDOs materials on the chloride ion permeation resistance of cementitious is conducted through macroscopic tests, microscopic analysis and numerical simulation results. The compressive strength and chloride ion concentration distribution of concrete doped with different amounts of LDOs were tested, and the adsorption mechanism of LDOs was characterized and analyzed by XRD and SEM microscopic means. In the end, the numerical simulation validated the test results. The main conclusions were as follows.

(1)The improvement of the compressive strength of concrete by incorporating hydrotalcite roasted at 500 °C was obvious. The compressive strength of hydrotalcite concrete increased first and then decreased with the increase in LDOs, and the optimal amount of LDHs was 2% wt. Its 28-day compressive strength was increased by 10.95% compared to the control group, meeting the requirement for design strength.(2)The one-dimensional accelerated chloride penetration test showed that the roasted LDOs have good chloride ion adsorption capacity; the chloride ion concentration increased firstly and then decreased with the increase in hydrotalcite dosages. In the chloride salt environment, chloride ions diffused from the surface to the interior of the concrete through the water, and the chloride ion concentration of the concrete in the REF group was much higher than that of the concrete mixed with hydrotalcite.(3)The mechanism of solidifying chloride ions in LDOs materials was investigated by means of XRD and SEM microscopic tests from two perspectives: physical phase composition and microscopic morphology. Compared with the LDHs material, the LDOs material treated by roasting at 500 °C appears to be a spinel material and does not possess the characteristic diffraction peaks of MgAl-CO_3_ LDHs. The LDOs material can complete the structural reconstruction by the adsorption of chloride ions after the hydration reaction in NaCl solution for 24 h, and its microscopic morphology is still obvious in the form of the lamellar structure.(4)The numerical simulation analysis by COMSOL shows that the simulation results of chloride ion concentration with diffusion depth are in good agreement with the experimental results, and the two conclusions are consistent. The amount of hydrotalcite has different effects on chloride ion diffusion, and according to the diffusion depth, it can be concluded that the anti-chloride ion permeation performance is in the following order: L2 group > L4 group > L6 group > REF group.

## Figures and Tables

**Figure 1 materials-16-06349-f001:**
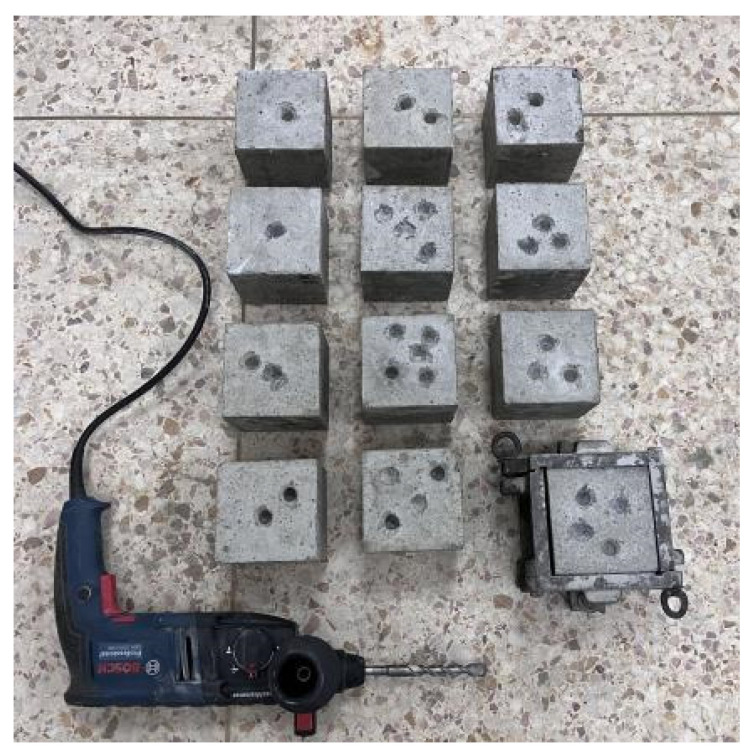
Test specimen sampling schematic.

**Figure 2 materials-16-06349-f002:**
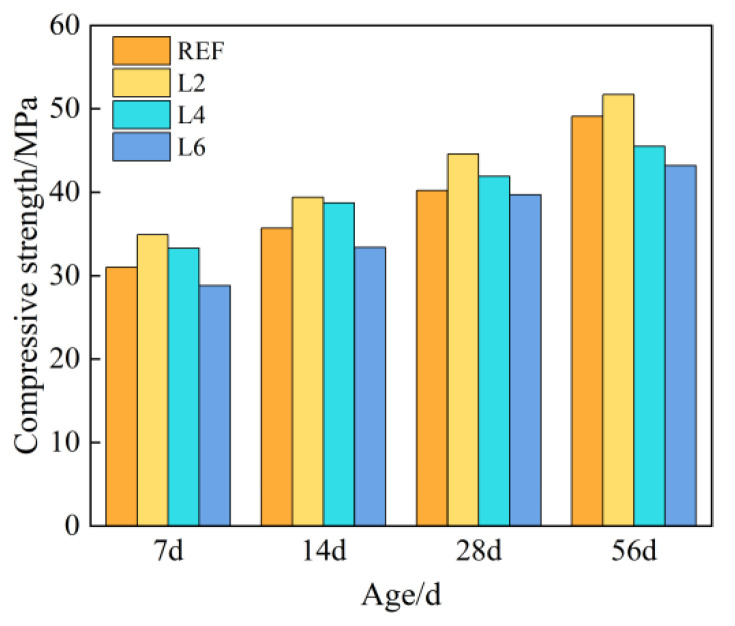
Compressive strength of concrete with different LDOs dosage.

**Figure 3 materials-16-06349-f003:**
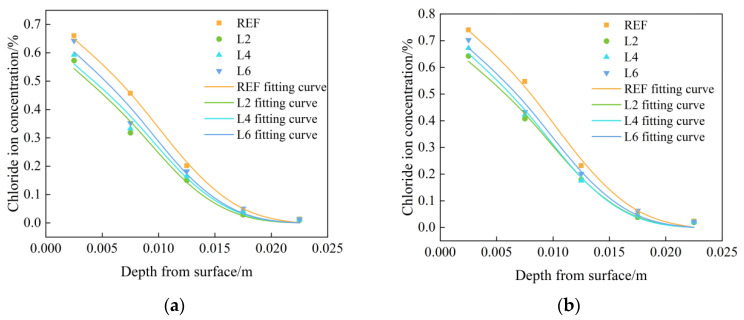
Chloride ion distribution test curve and fitting curve of specimens mixed with LDOs in chloride salt environment immersed for (**a**) 30 days and (**b**) 60 days.

**Figure 4 materials-16-06349-f004:**
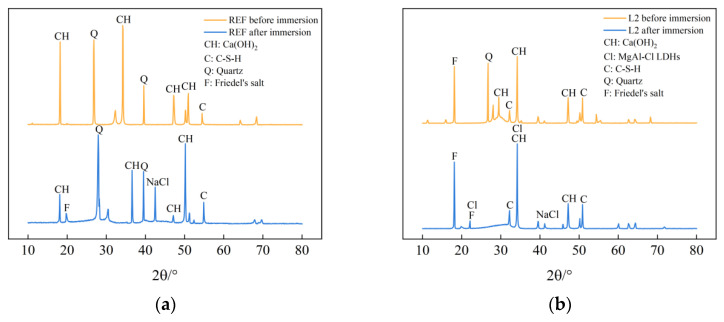
XRD diffraction spectra of specimens from (**a**) REF group and (**b**) L2 group before and after immersion in chlorine salt environment.

**Figure 5 materials-16-06349-f005:**
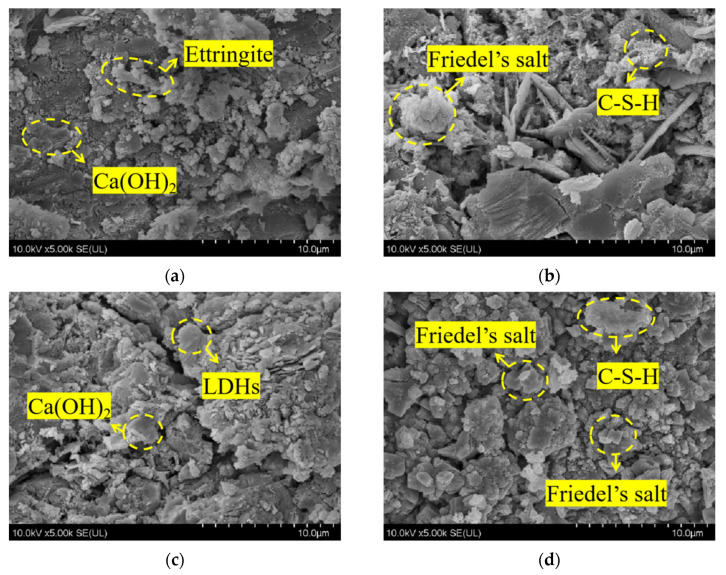
SEM microscopic morphology of (**a**) REF group before immersion, (**b**) REF group immersed for 56 days, (**c**) L2 group before immersion and (**d**) L2 group immersed for 56 days in chlorine salt environment.

**Figure 6 materials-16-06349-f006:**
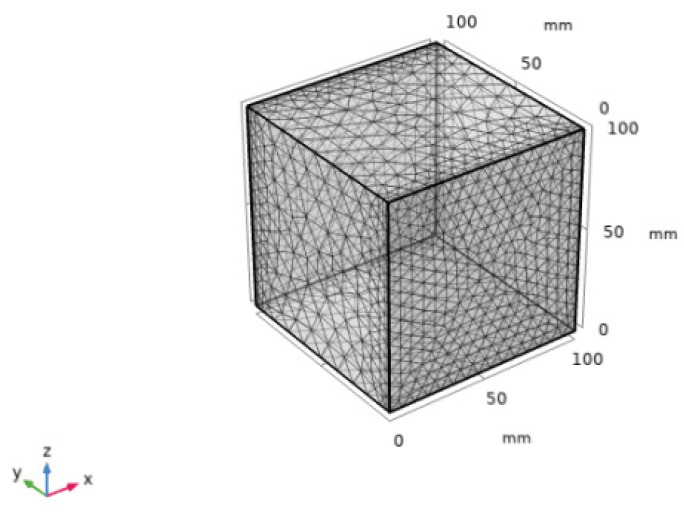
Three-dimensional concrete model mesh generation.

**Figure 7 materials-16-06349-f007:**
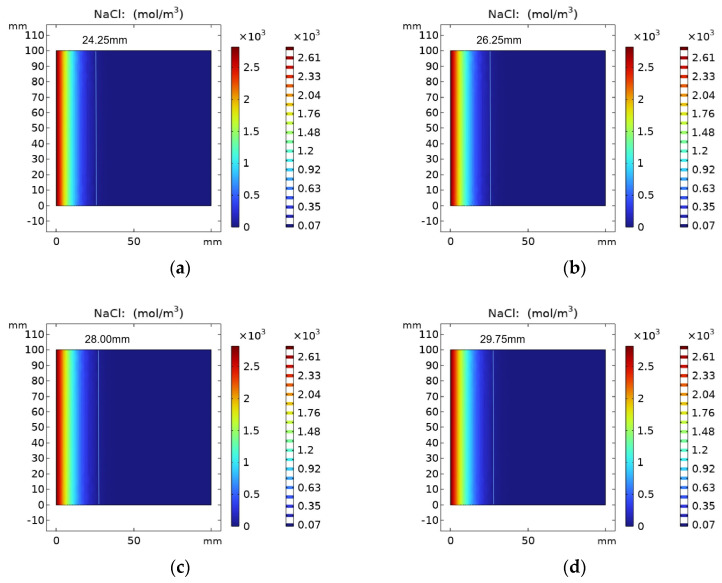
Chloride ion concentration distribution in REF group at (**a**) 30 days, (**b**) 60 days, (**c**) 90 days and (**d**) 120 days.

**Figure 8 materials-16-06349-f008:**
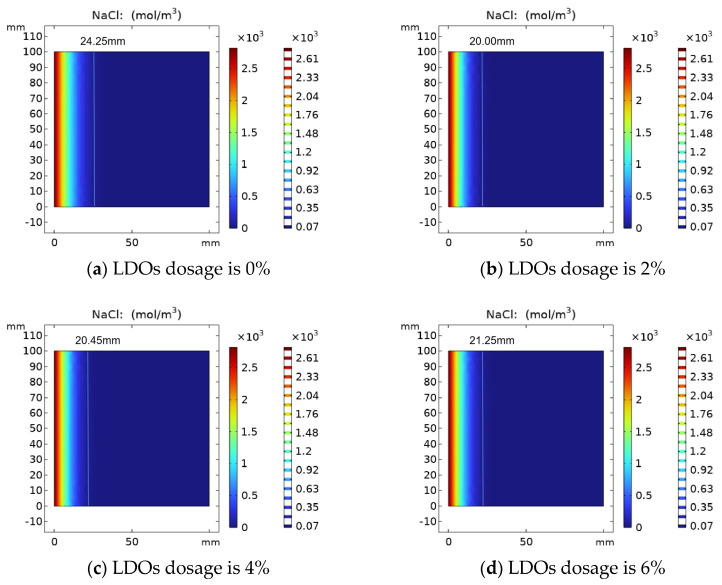
Variation of chloride ion concentration with (**a**) 0%, (**b**) 2%, (**c**) 4% and (**d**) 6% LDO dosage.

**Figure 9 materials-16-06349-f009:**
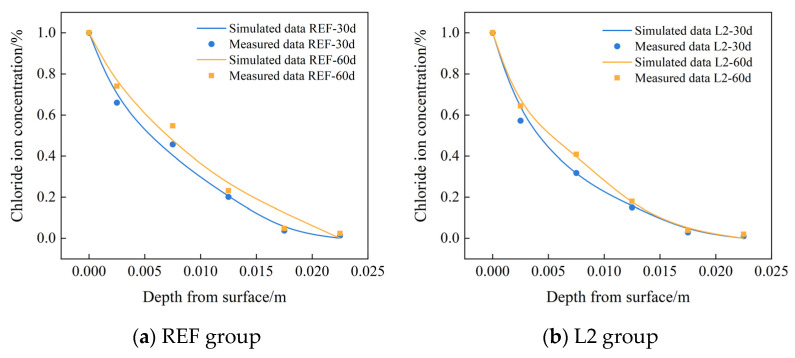
Comparison of COMSOL numerical chloride ion concentration at different depths with experimental data.

**Table 1 materials-16-06349-t001:** Chemical composition of cement and hydrotalcite (%).

Chemical Composition	SiO_2_	Al_2_O_3_	Fe_2_O_3_	CaO	MgO	SO_3_	LOI
LDHs	3.26	44.63	0.23	0.40	50.06	1.28	0.32
Cement	23.92	4.65	3.99	61.71	2.18	3.68	0.98

**Table 2 materials-16-06349-t002:** Mix proportions and properties of hydrotalcite concrete.

No.	Kg/m^3^					Compressive Strength (MPa)		30-Day Chloride Ion Permeability Coefficient (×10^−13^ m^2^/s)
	Cement	LDOs	Sand	Aggregate	Water	7 d	28 d	
REF	346.00	0	759	1138	156	31.2	40.2	7.35
L2	339.08	6.92	759	1138	156	34.8	44.9	5.91
L4	332.16	13.84	759	1138	156	33.9	43.0	7.20
L6	325.24	20.76	759	1138	156	28.8	39.8	8.88

**Table 3 materials-16-06349-t003:** Fitting results of chloride ion diffusion coefficient and surface chloride ion concentration under chloride salt environment.

Immersion Age/d	Specimen Number	Chloride Ion Diffusion Coefficient/×10^−13^ m^2^/s	Surface Chloride Ion Concentration/%	Correlation Coefficient
30	REF	7.35	0.67	0.99
	L2	5.91	0.57	0.98
	L4	7.20	0.58	0.97
	L6	8.88	0.62	0.97
60	REF	4.33	0.76	0.99
	L2	3.13	0.65	0.99
	L4	3.49	0.69	0.98
	L6	3.55	0.70	0.98

## Data Availability

Data will be made available on request.

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
