# Peer review of "Experimental Study and Numerical Analysis of Chloride Ion Diffusion in Hydrotalcite Concrete in Chloride Salt Environment"

_materials, 2023, doi:10.3390/ma16196349_

Round 1

Reviewer 1 Report

This work presents

Hydrotalcite, known as layered double hydroxides (LDHs), is a new type of admixture to delay the corrosion of reinforcement. The aim of this study was to investigate the chloride ion diffusion behavior of C30 concrete with varying amounts of calcined hydrotalcite (0%, 2%, 4% and 6%) under a chloride salt environment. NT-Build 443 test was adopted to characterize the one-dimensional accelerated chloride ion penetration of concrete. The distribution of chloride ion concentration in hydrotalcite concrete with different mix proportions immersed in sodium chloride solution for 30 days and 60 days was determined, and the chloride ion diffusion coefficient and surface chloride ion concentration were fitted based on Fick's second law to establish the chloride ion diffusion model considering the influence of multiple factors. This model was validated through COMSOL Multiphysics finite element software. The results show that the chloride ion diffusion process under chloride salt immersion is in accordance with Fick's second law. The resistance to chloride ion penetration of concrete with 2% calcined hydrotalcite is the best. The calculated chloride ion concentrations by COMSOL software and test results are in good agreement which verifies the reliability of chloride ion diffusion model.

1.     There are mistakes in subscript and Super subscript. For example, near Figure 7 7.35×10-13 m2/s. Check all such kinds of mistakes and revise them throughout the paper.

2.     Your paper explains steps for geometry, Physics, and mesh. How you get the results etc add more detail.

3.     Did you compare experimental and simulation results? What is percentage error?

4.     How you can do simulation without mesh independence test. How do you say the accuracy of results?

5.     Your samples for testing are in 3D and the model is in 3D then why showing only 2D surface plots?

6.     Did you use the same material properties in COMSOL and concentrate for the experiment?

7.     You did the same testing on COMSOL and experimental? Or just did the characterization of the material?

8.     How can you compare your results with existing literature to verify your work contribution or novelty?

9.     How can you link compression testing with the dilute simulation?

10.  Add the following research article as a reference related to erosion.

-Centrifugal compressor stall control by the application of engineered surface roughness on diffuser shroud using numerical simulations

Author Response

The authors wish to thank the editor and three reviewers for their very thorough, insightful, and constructive reviews. All comments have been addressed in the revised version of the manuscript. Original comments of each reviewer have been placed in black font in this document and specific responses have been placed in red font for ease of reference. In the responses, all the numbers of figures, tables, and lines refer to the revised manuscript, unless otherwise specified.

Reviewer 2 Report

I decided to reject the manuscript for the following reasons:

i) Section 2.7 describing the chloride ion transport is hard to follow and needs to be carefully rewritten. Several formulas are presented with a lot of symbols without any explanations. I am missing some idea/text why they are related together. Several typos and misleading symbols appear within this section.

ii) The same as above is valid for Section 4 summarizing the numerical results. The authors first describe the problem as a 3D, then present results only for two-dimensional cuts. It appears to me that the 1D simulation should be sufficient enough. The reason for spatial modeling is the heterogeneity of material or different loading conditions in each direction or something else. You should explain it in the text; otherwise, it is confusing. The scheme describing the initial and loading conditions is missing. 

iii) Multiple occurrences - the meaning of measured values presented in the manuscript is unclear. Do they represent average values? 

iv)  Multiple occurrences - Please, avoid the use of imperative sentences. 

Author Response

(The authors gave the same response as above.)

Reviewer 3 Report

The paper relates to the investigation of the chloride ion diffusion behaviour of C30 concrete with varying amounts of calcined hydrotalcite (0%, 2%, 4% and 6%) under a chloride salt environment. The model proposed was validated through COMSOL 17 Multiphysics finite element software. The achieved test results are in good agreement which verifies the reliability of the chloride ion diffusion model. They show that the chloride ion diffusion process under chloride salt immersion fulfils Fick's second law. The issue is interesting because it allows you to delay the corrosion of reinforcement in concrete element.

 I have a few comments:

1.Line 91: Layered double Hydrotalcite, and the abbreviation is LDH - capital letters

2.Line 410 at the sentence with … of was 2%...

3. A workchart is needed to present what reacts with what (assumptions and actions) and what the result is obtained, with accuracy and impact on the properties of the material.

4.Line 142. To what humidity were the samples dried?

5. In the tests, the strength of the samples, what value was applied to the load and (conditions of supporting the sample) and was it on the entire surface of the wall or only in one point?

6.The LDOs material could adsorb the chloride ions penetrated into the concrete pore interior by interlayer anion exchange ability under the chloride salt environment, and complete the structural reconstruction while improving the chloride salt attack resistance of concrete. This property achieved is very interesting and important. But, what about testing reinforced concrete elements, and not only concrete ones? - the influence of the chloride ion diffusion behaviour on reinforced concrete elements?

7. What about research on the impact of the chloride ion diffusion behaviour on steel loss?

sufficient for publication

Author Response

(The authors gave the same response as above.)

Round 2

Reviewer 1 Report

All of my concerns are properly addressed.

Author Response

The authors thank the reviewer for the valuable and careful comments.